# Fitness Fights Fires: Exploring the Relationship between Physical Fitness and Firefighter Ability

**DOI:** 10.3390/ijerph182211733

**Published:** 2021-11-09

**Authors:** Annmarie Chizewski, Allyson Box, Richard Kesler, Steven J. Petruzzello

**Affiliations:** 1Department of Sport and Exercise Science, College of Education, Benedictine University, Lisle, IL 60532, USA; 2Department of Kinesiology and Community Health, College of Applied Health Sciences, University of Illinois Urbana-Champaign, Champaign, IL 61820, USA; agbox2@illinois.edu (A.B.); rkesler2@illinois.edu (R.K.); petruzze@illinois.edu (S.J.P.); 3Illinois Fire Service Institute, Champaign, IL 61820, USA

**Keywords:** fitness, firefighters, health

## Abstract

*Background**:* Regular exercise in firefighters may be effective in preventing or attenuating ill health (e.g., hypertension, diabetes, and obesity), as well as improving their firefighting ability. The purpose of this study was to investigate the relationship between physical fitness and firefighting ability. *Methods*: Male firefighter recruits’ (*n* = 89; age = 27.1 ± 4.2 yrs) physical fitness and firefighting ability were assessed at Week 1 and Week 7 during a basic firefighting training academy. Physical fitness was assessed via 1.5 mile run time, sit-up and push-up repetitions, the Young Men’s Christian Association (YMCA) bench press test, vertical jump height, and sit-and-reach flexibility, while firefighting ability was assessed via completion time on a firefighting skills test. *Results:* Fitness predicted significant variance in firefighting ability at Week 1 (*R*^2^
*=* 0.46; *p* < 0.01) and Week 7 (*R*^2^ = 0.46; *p* < 0.01), after accounting for age and body mass index. Cardiovascular endurance accounted for 22.4% (*F*Δ (1, 85) = 25.75) and 39.3% (*F*Δ (1, 85) = 55.53) while muscular endurance accounted for an additional 19.0% (*F*Δ (3, 82) = 10.34) and 6.3% (*F*Δ (3, 82) = 3.2) unique variance in firefighting ability at Week 1 and Week 7, respectively. *Conclusions:* Given the strong association between fitness and firefighting performance, municipal departments may want to focus on increasing fitness levels among firefighters.

## 1. Introduction

Firefighting involves various aspects of physical fitness, including cardiovascular endurance, muscular strength and endurance, power, agility, and flexibility. Despite the physical strain firefighting places on the men and women who choose this profession, the vast majority of career and volunteer firefighters fail to maintain the needed levels of physical fitness to function safely and efficiently while on duty [1,2,3,4,5]. While they can perform the tasks necessary for the job, the additional strain experienced due to their lack of fitness could lead to deleterious health outcomes over time. There is a clear and apparent need for physical fitness and exercise standards in the fire service, which warrants more research to be done to help improve the health of firefighters [6,7].

Due to the mismatch of the physical demands of firefighting and the current physical fitness level of many in the fire services, cardiac incidents and over-exertion are the leading causes of on-duty deaths. Cardiac incidents alone account for 40–50% of on-duty death among firefighters [8]. While there are some non-modifiable risk factors that contribute to the likelihood of experiencing a cardiac incident (e.g., age, gender, family history, and environmental work-related risks), there are also many modifiable risk factors that fire departments can address to enhance and promote behavior changing strategies to reduce risk of a cardiac incidents. Some of these modifiable risk factors include obesity, high blood pressure, smoking status, poor nutrition, poor hydration, and lack of physical activity and physical fitness [8,9]. Indeed, recent research has begun to examine the relationship between physical fitness and the likelihood of experiencing a cardiac incident among firefighters and these findings suggest that many of the modifiable risk factors can be targeted to prevent cardiac incidents among firefighters [8,9,10,11,12].

Obesity rates among firefighters are alarmingly high and raise major health concerns among this population. Previous research has found that for every one-unit increase in body mass index (BMI), job disability increased 5% [7]. Firefighters with a BMI of ≥30.2 kg·m^2^ had a significantly increased risk of experiencing work-related injury when compared to firefighters whose BMI was <27.2 kg·m^2^. Further, previous studies have demonstrated unfavorable relationships between BMI and resting blood pressures, various health-related blood markers, and overall morbidity scores [10]. Given the high levels of obese firefighters and the undeniable adverse effects that such conditions can have on overall health, research regarding physical fitness and the implementation of physical activity programs seems warranted.

Promoting physical fitness and physical activity may be a cost-effective strategy fire departments can use to target many of these unfavorable risk factors. Furthermore, findings have shown that individuals who have higher BMIs and lower levels of physical fitness are more likely to experience injury while on duty and perform worse on work-related physical tasks when compared to their more fit counterparts [6,13,14,15,16,17]. Given that there is a clear need for adequate fitness levels to not only perform efficiently while on duty, but also to reduce the likelihood of sudden cardiac incidents, more research is warranted to determine the specific relationship between physical fitness and firefighting ability.

Due to the need for a better understanding of the relationship between physical fitness and firefighting ability, the present study examined this relationship at Week 1 of a Basic Firefighter Academy among newly hired recruit firefighters with little-to-no firefighting experience, as well as examining the same relationship at Week 7 of the training academy once firefighting techniques and skills had been learned and improved. Further, we wanted to explore if change in fitness impacted change in (Δ) firefighting ability. We hypothesized that cardiovascular endurance, muscular endurance, power, and flexibility would be related to firefighting ability, measured by performance time on the Academy Firefighting Challenge (AFC) at Week 1 and Week 7. We further hypothesized that greater increases in physical fitness over the course of a 7 week training program would be associated with a greater change in firefighting ability (i.e., shorter completion time on the AFC).

## 2. Materials and Methods

The present study was approved by a Midwestern university Institutional Review Board. The participants were male firefighter recruits (see Table 1 for Participant Characteristics and Table 2 for fitness characteristics) enrolled in the Basic Firefighting Academy training course held at a Midwest fire academy. The Basic Firefighting Academy is held biannually (i.e., Fall, Spring). Data for this project were collected during Spring 2018, Fall 2018, and Spring 2019. Please note, there was only one female recruit enrolled during those periods, thus her data were excluded from all data analysis. There were no significant differences in participant characteristics (age; *df* = 2, *F* = 0.70, *p*= 0.5, weight; *df* = 2, *F* = 0.44, *p* = 0.65, height; *df* = 2, *F* = 2.38, *p* = 0.10, and BMI; *df* = 2, *F* = 0.33, *p* = 0.98) between the three recruit class cohorts. Further, at Week 1 all three cohorts had an average estimated VO_2max_ that was classified as “poor” according to the American College of Sports Medicine Cardiorespiratory Fitness Classifications [18].

During the first three days of both the first (Week 1) and final (Week 7) weeks of each academy course, firefighter recruits completed fitness and firefighting ability testing. All testing was supervised by academy instructors and researchers to assure quality repetitions and safety. Trained researchers administered and scored all testing at Week 1 and Week 7. All test administrators went through training to assure each test was administered accurately. The same researchers administered each test for the same participants during Week 1 and 7. Note, testing was completed across three days to reduce the potential for acute muscular fatigue. In addition, all testing sessions were conducted at 0630 h.

Day 1. The recruits were instructed to complete as many push-ups as possible in 60 s. Their partner kept track of the number of correctly completed repetitions and then recorded the score. A correct repetition required the recruit to bend their elbow and lower their entire body as a single unit until their upper arms were at least parallel to the ground. They then returned to the starting position by raising their entire body until their arms were fully extended. The only acceptable resting position was upward into a pike. The test was terminated if any portion of their body (e.g., stomach and knee) touched the floor, and only repetitions completed prior to that occurring were reported. This format of testing was the standard used in previous academies and was adopted for testing purposes during data collection.

The recruits were also instructed to complete as many sit-ups as possible in 60 s. They were instructed to assume the starting position by lying on their back with their knees bent at a 90-degree angle. Their feet were allowed to be together or up to 12 inches apart while another person held their ankles with only their hands. No other method of bracing or holding the feet was authorized. The heel was the only part of their foot that had to stay in contact with the ground and their fingers had to remain interlocked behind their head. On the command “Go”, they began raising their upper body forward to, or beyond, the vertical position (i.e., the base of their neck was above the base of their spine). After they reached or surpassed the vertical position, they lowered their upper body until the bottom of their shoulder blades touched the ground. Their head, hands, arms, or elbows did not have to touch the ground. Repetitions did not count if they failed to reach the vertical position, failed to keep their fingers interlocked behind their head, arched or bowed their back, raised their buttocks off the ground, or let their knees exceed a 90-degree angle. This format of testing was the standard used in previous academies and was adopted for testing purposes during data collection.

Finally, the recruits were instructed to complete a 1.5 mile course as quickly as possible, ideally by running. Time to complete the 1.5 miles was recorded by research staff to the nearest second. The outdoor course was laid out on the grounds of the Academy. Estimated aerobic capacity was determined using the formula: (3.5 + (483·× 1.5 mi·run time−1)) [18].

Day 2. Weight (body mass, in kg) and height (without shoes, in cm) were measured on a Seca 284 digital scale. Body mass index (BMI) was calculated as body weight divided by height in meters squared (kg·m^2^) [18].

The YMCA bench press protocol followed procedures of Golding et al. [19]. Participants were asked to complete as many repetitions as possible to the beat of a metronome set to 60 b∙min^−1^, with the press up or the return to the chest occurring with each click of the metronome, or until they reached a maximum of 60 repetitions. The total weight of the barbell was 80 lbs (36 kgs) for males. The test was terminated if unable to maintain the pace of the metronome. Participants were given 4-metronome beats (i.e., 2 full repetitions) to correct their pace before termination. 

Hamstring and lower back (trunk) flexibility was assessed using a sit-and-reach box [19]. The participants sat on the floor with legs stretched out, backs of the knees flat on the floor, and soles of the feet (without shoes) flat against the back of the sit-and-reach box. With hands placed one on top of the other, they were instructed to push a slide as far forward as possible in one smooth motion. Distance reached was recorded to the nearest centimeter. Participants were allowed three attempts and the best of the three attempts was recorded.

Finally, to assess power, the participant stood near a Jump USA Vertec Vertical Jump System with one arm fully extended so research staff could record their standing height. The participant then jumped up and touch the highest possible vane of the Vertec System. The jump height was the difference between standing height and jumping height and is used as an indicator of power [20]. Participants were allowed three attempts and the best of the three attempts was recorded.

Day 3. Firefighting ability was assessed at Week 1 and Week 7 via the Academy Firefighting Challenge (AFC). The AFC is a six-event physical performance test used to assess cardio-respiratory fitness and muscular endurance. This is an academy-specific test patterned after the Candidate Physical Ability Test [21], with the various tasks designed to mimic what a firefighter would encounter in everyday situations while on duty. The firefighters wore full turnout gear during the AFC, including helmet, bunker pants, bunker coat, boots, gloves, and self-contained breathing apparatus (SCBA; consists of facemask, regulator and hose, and air pack harness and air bottle) equipment. The total weight of the gear was approximately 45 pounds. The AFC consisted of the following events: Forcible Entry, Search-Crawl, Victim Drag, Hose Advance, Equipment Carry, and Ladder Raise and Extension. A research staff member recorded the lap splits (time for each event) and overall time, while also directing the participant between stations. 

Station 1: Forcible Entry. The forcible entry event required the use of a 10 pound (4.54 kg) sledgehammer to strike the measuring device (Keiser^®^ Sled) until the sled was moved a distance of 5 feet. There was a 90 s time cap for this event. That is, if the participant was unable to move the sled the required distance in 90 s they were instructed to stop and move on to the next station. Please note, no recruits reached the cap during the event (i.e., all completed in <90 s).

Station 2: Search-Crawl. The firefighters entered the SCBA “can”, a 40 ft × 8 ft sea-land container modified specifically for this task and crawled through the course on hands and knees. The SCBA crawl was a U-shaped maze that had low visibility and a floor that was uneven, resulting in variable ceiling heights. No obstacles were present through the course.

Station 3: Victim Drag. Participants were required to carry or drag a 110 pound (49.90 kg) mannequin 100 ft to a pre-marked end point. They were allowed to grasp the mannequin in any manner they preferred. 

Station 4: Hose Advance. This event required the FF to grasp a 100 ft (30 m) 1¾ inch (44-mm) charged hose and drag the hose 75 feet to a prepositioned cone. They were allowed to place the hose line over their shoulders or across their chest, as long as no more than 8 ft of the hose was placed over their body.

Station 5: Equipment Carry. The recruits picked up two saws (~15 lbs per saw), one in each hand, and carried them 50 ft to a designated turn-around point where firefighters then turned around and carried the equipment 50 ft back to the starting line. They were permitted to place the saw(s) on the ground and adjust their grip as needed. Once they returned to the starting line, they placed each saw on the ground and moved on to the final station. 

Station 6: Ladder Raise and Extension. The recruits approached a prepositioned 28 ft two-section fixed ladder (i.e., attached at the bottom) that was lying flat on the ground. They were instructed to raise the ladder as quickly as possible from the ground to a fixed vertical position (90 degrees) from its initial position. 

In addition to the fitness testing, the firefighter recruits participated in a 7 week fitness program while enrolled in the Academy. Firefighter recruits reported to the Academy campus at 06:30 every weekday morning of the 7 week Academy for daily physical training (PT). Daily PT began with a dynamic warm-up (which took approximately 10 min). This included jumping jacks, jump rope, and dynamic stretching. Following the warmup, the recruits were led through approximately 40 min of high-intensity functional training (HIFT) [22]. The activities varied daily, and highlighted aspects of fitness required for safe and efficient firefighting: aerobic capacity, muscular strength and endurance, power, flexibility, and agility. The program gradually incorporated movements and equipment commonly used during firefighting tasks (e.g., hoses, sledgehammer). The PT sessions were led and supervised by Illinois Fire Service Institute (IFSI) staff members. The PT schedule was designed by an exercise specialist and firefighter expert who consulted with a certified CrossFit^®^ coach. The HIFT program varied daily and incorporated exercises such as stair climbs, bear crawls, hose drags, body weight circuits, box step-ups, ability group runs, and flexibility/mobility. In addition to daily PT, the recruits also learned and practiced a multitude of firefighting skills during their time at the Academy. Daily activities included classroom work and written exams for part of the day as well as fire-ground training. The fire-ground training included full turn-out gear, practice with fire-ground tools (i.e., sledgehammers, hoses, ladder, etc.), and live fire drills which were all supervised by the IFSI staff members. Recruits were on campus for approximately 12 h a day from Monday to Friday.

### Data Processing and Analysis

Descriptive statistics were calculated for each of the variables under consideration. Statistical analyses were conducted using SPPS 24.0 for Windows (SPSS, Chicago, IL, USA). A significance level of alpha ≤0.05 was chosen to denote statistical significance. Bivariate correlations were used to determine the magnitude and direction of the relationship between various aspects of fitness (cardiovascular endurance, muscular endurance, flexibility, and lower body power) and firefighting ability (Keiser sled, SCBA crawl, victim drag, hose advance, equipment carry, ladder raise, and total completion time) at Week 1 and Week 7. To explore the relationship between fitness and firefighting ability further, a hierarchal regression was conducted to examine unique variance of each fitness variable. Such analysis allows researchers to determine if certain aspects of fitness impacted firefighter ability more than others. For all regressions, Age and BMI, while not significantly correlated to firefighting ability in this sample, were retained as predictors due to previous evidence suggesting their inverse impact on exertion task ability [2]. Given previous literature highlighting the importance of cardiovascular fitness in firefighters [8,9,10,11,12], estimated VO_2max_ was entered at the second level, followed by muscular endurance. It is important to note that the three indices of muscular endurance (i.e., YMCA bench press, 60 s sit-up and push-ups) were retained as only moderate relationships existed among each of the variables within this data set (see Table 3 and Table 4). Lastly, power assessed via vertical jump was entered as the final level of the regression and flexibility was excluded to the non-significant relationships with firefighting ability.

While the primary intent of this project was to examine the relationship between fitness and firefighting ability during the first and final weeks of a basic training academy, it was also of interest to determine if change in overall fitness significantly impacted change in firefighting ability. That is, were improvements in fitness associated with improvements in firefighting ability. To inspect this relationship, we created a Δfitness variable. This variable was a composite score of the following variables: estimated VO_2max_ (mL·kg^−1^·min^−1^), YMCA bench press (reps), push-ups (reps), sit-ups (reps), flexibility (cm), vertical jump (in). Fitness changes were calculated for each fitness variable by subtracting Week 1 score from Week 7 score, and then these separate change scores were summed together to determine ΔFitness. In addition, a ΔFFAbility score was calculated following a similar procedure. That is, a change score of time to complete the firefighting ability tasks was calculated by subtracting Week 7 time from Week 1 time which resulted in ΔFFAbility score.

## 3. Results

Bivariate correlations (Table 5 and Table 6) revealed significant relationships between cardiovascular endurance (*r* = −0.49, *p* ≤ 0.01), bench press (*r* = −0.51, *p*≤ 0.01), push-ups (*r* = −0.38, *p* ≤ 0.01), sit-ups (*r* = −0.41, *p* ≤ 0.01), power (*r* = −0.32, *p* ≤ 0.01) and total firefighting ability (total completion time) at Week 1. At Week 7, significant relationships were revealed between cardiovascular endurance (*r* = −0.53, *p* ≤ 0.01), bench press (*r* = −0.40, *p* ≤ 0.01), and sit-ups (*r* = −0.27, *p* ≤ 0.05) and total firefighting ability (total completion time). Age, BMI, and Flexibility were not significantly related to overall firefighting ability during Week 1 or 7. In addition, push-ups and power were not associated with firefighting ability during Week 7.

We also examined the relationship between BMI and fitness at Weeks 1 and 7. Simple bivariate correlations revealed that significant relationships with all aspects of physical fitness except for flexibility (Table 7 and Table 8). Correlations also revealed that BMI at Week 7 was not significantly correlated to change in fitness and change in firefighter ability over the course of the 7-week academy.

Hierarchal linear regressions (Table 9 and Table 10) were conducted to determine the unique variance explained beyond Age and BMI for each fitness variable at Week 1 and Week 7. At Week 1, after accounting for age and BMI (3.4% variance, *p* ≥ 0.05), cardiovascular endurance, muscular endurance, and power predicted an additional 42.3% variance in FF ability, with cardiovascular endurance accounting for 22.4% (*F*Δ = (1, 85) = 25.75; *p* < 0.001) unique variance and muscular endurance accounting for an additional 19.0% (*F*Δ = (3, 82) = 9.43; *p* < 0.01) unique variance. At Week 7, after accounting for age and BMI (<1.0% variance, *p* > 0.05), fitness (cardiovascular endurance, muscular endurance, and power) predicted an additional 46.1% variance in FF ability, with cardiovascular endurance accounting for 39.3% (*F*Δ = (1, 85) = 55.53 *p* < 0.001) unique variance and muscular endurance accounting for an additional 6.3% (*F*Δ = (3, 82) = 9.43; *p* < 0.05) unique variance. Power did not account for any additional variance at Week 1 or 7.

An additional hierarchal regression (Table 11) was conducted to determine if Fitness predicted unique variance in Performance Week 7. At Week 7, after accounting for age and BMI (6% variance, *p* = 0.06), Fitness significantly accounted for 6.0% (*p* = 0.02) unique variance on FFAbility.

## 4. Discussion

The present study sought to determine the extent performance during a physical ability challenge test (i.e., AFC) was related to various components of physical fitness in recruit firefighters. We hypothesized that firefighter recruits who possessed greater levels of physical fitness would perform better on the AFC. We also sought to explore the extent to which each component of fitness predicted performance on the AFC and determine if change in fitness predicted Week 7 AFC performance. Our results revealed that better physical fitness was associated with better performance on the AFC. Specifically, cardiovascular endurance (i.e., estimated aerobic capacity based on 1.5 mi·run time^−1^), muscular endurance (i.e., bench press, push-ups, and sit-ups repetitions), and power (vertical jump) were significantly correlated with firefighting ability assessed via the AFC during Week 1. Similar patterns emerged during Week 7, with cardiovascular endurance and muscular endurance significantly correlated with performance on the AFC. Additionally, our results revealed that components of fitness accounted for ~41% and ~46% total variance in performance on the AFC at Week 1 and Week 7, respectively, after controlling for age and BMI. Cardiorespiratory endurance accounted for the majority of this variance, while muscular endurance accounted for the remainder of the explained variance (~19%, ~6%) at Week 1 and Week 7, respectively. While power was significantly correlated with firefighting ability during Week 1, it did not predict any variance in firefighting ability beyond that explained by cardiorespiratory and muscular endurance. Contrary to our hypothesis, flexibility was not associated with performance on the AFC. Additionally, we found that the change in fitness accounted for a significant amount of unique variance (6%) on change in firefighting ability assessed via performance on the AFC at Week 7. These findings support previous research that highlights the importance of physical fitness on the ability to perform firefighting tasks efficiently, as indexed by faster completion of the various tasks of the AFC in those with greater aerobic capacity and muscular endurance. 

When firefighters are engaged in fire-ground activities, while wearing heavy and cumbersome personal protective equipment, a significant burden can be placed on the cardiovascular system [4,9,10,22,23]. In addition to cardiovascular strain, fire-ground activities (e.g., forcible entry, hose hoist, search/rescue, ceiling overhaul) can require varying degrees of muscular endurance, strength and power. Research has found that firefighters who possessed lower levels of fitness were less likely to be able to complete two successive bouts of simulated firefighting work cycles [15]. Thirty firefighters (29 males, 1 female; average age = 30.4 ± 1.5 yrs; BMI = 27.4 ± 0.7 kg/m^2^; average VO_2max_ = 43.7 ± 1.3 mL·kg^−1^·min^−1^) engaged in 3 conditions: (1) one bout of firefighter tasks in an environmental chamber; (2) two bouts of firefighter tasks with rest outside the chamber; and (3) back-to-back bouts in the environmental chamber. Eleven of the 30 subjects failed to complete at least one of the two-bout activities because they felt too tired, too hot, too nauseous, or felt unsafe. These participants had lower levels of fitness, were heavier, and had higher BMIs than the 19 participants who successfully completed all work cycles [15]. They found that greater aerobic capacity was associated with more successful performance on simulated firefighting tasks. Our study was one of the first studies to explore which specific components of physical fitness are linked to better performance outcomes on firefighting tasks assessed via the AFC. Cardiovascular endurance (i.e., estimated VO_2max_) and muscular endurance (i.e., bench press repetitions, push-up repetitions, and sit-up repetitions) were significantly related to overall AFC completion time, as well as many of the specific firefighting skills assessed (i.e., Keiser^®^ sled, SCBA crawl, victim drag, hose advance, equipment carry, and ladder raise). These findings support the idea that sufficient fitness is needed to efficiently perform firefighting-specific tasks [4,9,15]. Combined, the present study and previous research suggest that individuals who are more physically fit are more capable of performing firefighting tasks when compared to their less-fit peers. 

Our AFC was designed to model the Candidate Physical Ability Test (CPAT [21]), a well-established and verified means of assessing firefighting ability. The CPAT test includes a stair mill climb, ladder raise and extension, forcible entry, search, rescue, and ceiling breach and pull. Researchers examined the relationship between fitness and performance on the CPAT in 33 career and volunteer firefighters. Their aerobic capacity (absolute VO_2_), anaerobic fitness (Wingate anaerobic cycling test), and firefighting ability (CPAT) were assessed, with the results indicating that anaerobic and cardiovascular fitness were the best predictors of overall CPAT performance [24]. Additionally, 57 (23 females) firefighters attempted the CPAT and only 91% of males and 15% of females were able to successfully complete the CPAT under the criterion time (10 min 20 sec) [25]. It was shown that relative VO_2_, body mass, and handgrip strength accounted for 67% of variance on CPAT performance. Both of these studies indicate that fitness is correlated to better performance on firefighting tasks, with cardiovascular endurance being a strong indicator of performance [24,25]. Similar to these studies, our results revealed that fitness accounted for significant variance on AFC performance. Specifically, cardiovascular endurance was significantly correlated to better performance on the AFC and accounted for a larger amount of significant variance on AFC performance than muscular endurance (~6–19%). Taken together, these previous and present findings suggest that greater cardiovascular endurance is strongly associated with performance on firefighting-specific tasks, but muscular endurance and strength should also be important indicators of firefighting ability. Given the strong association between fitness and firefighting performance, municipal departments should focus on increasing current physical fitness levels of firefighters to improve job performance. 

We also examined whether the change in fitness was associated with change in firefighting ability at Week 7. The improvement in fitness did predict significant variance (6%) in performance on the AFC at Week 7. Our findings suggest that greater improvements in fitness are associated with greater improvements on firefighting-specific tasks. Additionally, our findings indicate that changes in fitness can occur within 7 weeks and these changes can improve job-related performance in a relatively short amount of time.

We also found a moderate, yet unsurprising, relationship between BMI and physical fitness at Week 1 and Week 7. Individuals who possessed lower BMI scores had higher estimated VO_2max_ scores, were able to complete more push-ups and sit-ups, and scored better on the vertical jump test when compared to their peers who had higher BMI scores at Week 1 and Week 7. Alternatively, individuals who had higher BMI scores were able to complete more successful repetitions on the YMCA bench press test when compared to those with lower BMI score at Week 1 and Week 7. This could be explained by the nature of the tests used; the 1.5 mile run, 60 s push-up, 60 s sit-up, and vertical jump test are all body weight exercises so individuals with lower BMI scores had less mass to move and therefore were more successful. The YMCA bench press test requires individuals to successfully move an 80lb barbell, which may favor individuals with more mass. These findings support previous research which has found that firefighters who are considered overweight or obese have diminished health, generally possessed lower levels of fitness, and perform more poorly on physically demanding tasks when compared to their healthy/normal weight counterparts [1,2,3,4,8,13,26,27,28,29,30].

While the effects of fitness on fatigue were not studied during the current study, it is plausible that physical fitness may also act as a buffer against physiological fatigue while performing fire-ground activities. For example, previous research examined performance on a simulated fire-ground test (SFGT) in 12 trained (VO_2peak_ = 45.6 ± 3.3 mL × kg^−1^ × min^−1^) and 37 untrained firefighters (VO_2peak_ = 40.2 ± 5.2 mL × kg^−1^ × min^−1^) [14]. Dennison et al. found that when the trained firefighters performed the SFGT immediately following an exercise session, 70% of the trained firefighters completed the SFGT faster than the untrained firefighters [14]. These findings suggest physical fitness may act as a buffer against fatigue and/or improve recovery time following physically demanding tasks. 

One limitation of the present study was the lack of control or comparison group, as we utilized a pre-to-post-test design to test our hypothesis. As all participants simultaneously participated in a 7 week basic firefighting training academy, it is unknown whether improvements in firefighting ability were primarily due to fitness or, perhaps, were a result of the specific firefighting training tasks the recruits were required to undergo as part of the Academy. That is, the recruits were practicing the skills and drills seen during AFC on a regular basis during their time in the Academy. In addition, to the skills and drills the recruits practiced they also participated in daily HIFT which was designed to incorporate movements and equipment used on the fire ground in a progressive manner. As the academy progressed more fire-ground equipment such as hoses, sledgehammers, tires, litter carries, and dummy drags were incorporated into the HIFT program.

In addition, while several components of fitness were assessed, we were unable to conduct gold-standard muscular strength testing (i.e., 1 RM bench press, and 1 RM back squat) due to the constraints of the academy programming. It is very likely that muscular strength also plays a role in firefighting ability, like that of cardiorespiratory and muscular endurance. While we did not directly measure muscular strength, research has shown the the YMCA bench press test to be a reliable test that can be used to predict 1RM bench press for men and women [31]. When administering physical fitness testing in large populations, time and equipment can be restrictive factors. During our testing we needed to test 25–40 firefighters in 60 min and had limited equipment. Administering the gold-standard 1 RM bench press test was not feasible as it requires more equipment, more spotters, and can take up to 15 min per participant to complete [31].

Another potential limitation of the present study is that we did not control for nutrition, hydration status, or past work experience (e.g., manual labor, military experience, and construction). All three of these factors (i.e., nutrition, hydration, experience) may account for variance in firefighting ability and could allow researchers to better understand what contributes to enhanced firefighting ability. Poor nutrition and hydration have been linked to poor physical health and are often associated with decrements in physical performance [28,29,30]. Additionally, individuals who have experience in professions that require manual labor or who engage in physically demanding tasks (e.g., chopping, hammer or axe swinging, equipment carry, push, and pull movements, etc.) may be more inclined to perform better on fire-ground activities due to the similarities in required tasks.

Studying recruit firefighters to determine the relationship between physical fitness and firefighting ability is important and allows novice firefighters to better understand the importance of physical fitness as they embark on their fire service careers. However, future research should examine the relationship between fitness and firefighting ability in a more advanced, experienced sample where firefighting skills have been established. This is also of interest as previous research has suggested that firefighters are at an increased risk of experiencing a sudden cardiac incident as they age [5,9,32]. Thus, examining the relationship between fitness and firefighting ability in older and more experienced career firefighters is equally as important as studying this relationship in young recruit firefighters.

Additionally, we did not find any significant relationships between flexibility and firefighting ability; however, that does not mean flexibility should be neglected among this population. Close to half of the injuries experienced by firefighters are musculoskeletal-related injuries [33,34]. Poor movement quality and ability may be related to an increased risk of experiencing a musculoskeletal injury. By targeting flexibility early on in a firefighter’s career, injuries may be prevented which can lead to longer careers and healthier firefighters. One tool that may prove beneficial to municipal departments is The Functional Movement Screen [33]. This tool can be used to assess the effectiveness that physical training programs have on flexibility and movement quality among firefighters and other tactical athletes.

While the present study assessed firefighting ability via performance time on the AFC, we did not assess the technique and form executed during the AFC. Proper technique and form during fire-ground activities is important for safe, efficient firefighting and can reduce the risk of experiencing injury during demanding tasks. Examining whether physical fitness is associated with proper form and technique during simulated fire-ground activities would be another potential avenue of research. Such research can shed light on the impact physical fitness has on efficiency and safety during firefighting-specific tasks.

## 5. Conclusions

The present study provides evidence that multiple components of physical fitness are associated with better (i.e., faster) performance on simulated fire-ground activities. The findings suggest that cardiovascular endurance and muscular endurance were the strongest predictors for completing such tasks quickly. Given that the majority of career and volunteer firefighters are classified as overweight or obese, this information could be used by municipal departments, physicians, researchers, and exercise specialists to develop physical fitness standards and codes of conduct in the fire service to improve the quality of life and work performance of all firefighters. More specifically, by prescribing safe, effective, and relevant exercise, municipal departments can create healthier, safer, and more efficient firefighters. Exercise has been widely accepted a powerful tool to combat obesity and health-related issues that often arise as a side effect of physical inactivity.

## Figures and Tables

**Table 1 ijerph-18-11733-t001:** Participant Descriptive Statistics at Weeks 1 and 7.

Variables	Week 1	Week 7
Sample (*n*)	89	89
Age (years)	26.8 ± 4.2	-
Body Mass (kg)	89.24 ± 16.33	88.60 ± 15.15 **
Height (m)	1.78 ± 0.07	-
BMI (kg·m^2^)	28.11 ± 4.19	27.92 ± 3.82 **
Underweight (<18.5)	-	-
Normal (18.5–24.9)	22.5%	16.9%
Overweight (25.0–29.9)	43.8%	55.1%
Obese (>30)	33.7%	28.0%
Cardiorespiratory Fitness ^†^	40.84 ± 5.09	45.30 ± 5.24 **
Very poor *	11.2%	3.4%
Poor *	25.8%	6.7%
Fair *	51.7%	55.1%
Good *	10.1%	23.8%
Excellent *	1.1%	9.0%
Superior *	-	-

*Note.* ^†^ Estimated VO_2max_; * American College of Sports Medicine (ACSM) Cardiorespiratory Fitness Classifications [18]. ** *p*-value ≤ 0.001 when comparing Week 1 to Week 7.

**Table 2 ijerph-18-11733-t002:** Participant Fitness and Firefighting Ability Statistics at Weeks 1 and 7.

Variables	Week 1	Week 7
Sample (*n*)	89	89
1.5 Mile Run (min.s)	13.1 ± 1.8	11.7 ± 1.5 **
Push-Ups (reps)	41.9 ± 12.4	45.3 ± 5.2 **
Sit-Ups (reps)	31.4 ± 6.1	38.3 ± 7.8 **
Bench Press (reps)	30.4 ± 11.6	35.6 ± 11.6 **
Flexibility (cm)	7.6 ± 7.2	9.8 ± 7.1 **
Vertical Jump (in)	24.3 ± 3.7	24.4 ± 4.1
Kiser Sled (s)	44.3 ± 17.3	35.2 ± 8.9 **
SCBA Crawl (s)	44.2 ± 11.7	35.2 ± 8.9 **
Victim Drag (s)	22.5 ± 5.9	19.4 ± 4.6 **
Hose Advance (s)	15.2 ± 3.7	13.9 ± 3.7 *
Equipment Carry (s)	20.9 ± 3.2	19.3 ± 3.1 **
Ladder Raise (s)	7.4 ± 2.2	6.5 ± 1.5 **
Challenge Total (s)	240.2 ± 41.2	192.4 ± 41.6 **

*Note.* * *p*-value ≤ 0.05, ** *p*-value ≤ 0.001 when comparing Week 1 to Week 7.

**Table 3 ijerph-18-11733-t003:** Correlations between Muscular Endurance Variables at Week 1.

Variables	Push-Ups (reps)	Sit-Ups (reps)	Bench Press (reps)
Push-ups (reps)	-	0.545 **	0.450 **
Sit-ups (reps)	-	-	0.360 **

** *p* ≤ 0.01 level (2 tailed).

**Table 4 ijerph-18-11733-t004:** Correlations between Muscular Endurance Variables at Week 7.

Variables	Push-Ups (reps)	Sit-Ups (reps)	Bench Press (reps)
Push-ups (reps)	-	0.418 **	0.353 **
Sit-ups (reps)	-	-	0.381 **

** *p* ≤ 0.01 level (2 tailed).

**Table 5 ijerph-18-11733-t005:** Correlations between Fitness and Firefighting Ability at Week 1.

Variables	Keiser Sled (s)	SCBA Crawl (s)	Victim Drag (s)	Hose Advance (s)	Equipment Carry (s)	Ladder Raise (s)	Total Challenge Completion Time (s)
Age (yrs)	0.054	0.359 **	−0.034	0.000	−0.114	−0.051	0.099
BMI	−0.084	0.379 **	−0.006	0.121	0.052	−0.156	0.171
Estimated VO_2max_ (mL·kg^−1^·min^−1^)	−0.184	−0.530 **	−0.342 **	−0.266 **	−0.361 **	−0.044	−0.485 **
Push-Ups (reps)	−0.290 **	−0.361 **	−0.231 *	−0.289 **	−0.151	−0.258 *	−0.380 **
Sit-Ups (reps)	−0.259 *	−0.299 **	−0.169	−0.299 **	−0.363 **	−0.117	−0.407 **
Bench Press (reps)	−0.367 **	−0.308 **	−0.380 **	−0.328 **	−0.279 **	−0.301 **	−0.507 **
Sit and Reach (cm)	−0.107	−0.118	−0.080	0.055	0.033	−0.076	−0.096
Power ^†^ (in)	−0.243 *	−0.285 **	−0.206	−0.217 *	−0.251 *	−0.222 *	−0.318 **

* *p* ≤ 0.05 level (2 tailed); ** *p* ≤ 0.01 level (2 tailed). **^†^** Power was assessed via vertical jump.

**Table 6 ijerph-18-11733-t006:** Correlations between Fitness and Firefighting Ability at Week 7.

Variables	Keiser Sled (s)	SCBA Crawl (s)	Victim Drag (s)	Hose Advance (s)	Equipment Carry (s)	Ladder Raise (s)	Total Challenge Completion Time (s)
Age (yrs)	−0.161	0.039	−0.019	0.052	0.002	−0.006	−0.058
BMI	−0.113	0.276 **	−0.002	−0.272 **	0.112	−0.274 **	0.029
Estimated VO_2max_ (mL·kg^−1^·min^−1^)	−0.233*	−0.599 **	−0.404 **	−0.135	−0.520 **	−0.122	−0.526 **
Push-Ups (reps)	−0.176	−0.241 **	−0.227 *	−0.044	−0.299 **	−0.078	−0.165
Sit-Ups (reps)	−0.210 *	−0.228 *	−0.198	−0.098	−0.346 **	−0.165	−0.267 *
Bench Press (reps)	−0.408 **	−0.208	−0.411 **	−0.421 **	−0.408 **	−0.394 **	−0.400 **
Sit and Reach (cm)	−0.008	−0.135	−0.075	0.153	−0.009	0.108	−0.012
Power ^†^ (in)	−0.211 *	−0.098	−0.247 *	−0.131	−0.231 *	−0.193	−0.178

* *p* ≤ 0.05 level (2 tailed); ** *p* ≤ 0.01 level (2 tailed). **^†^** Power was assessed via vertical jump.

**Table 7 ijerph-18-11733-t007:** Relationship between Fitness and BMI at Week 1.

Variables	Estimated VO_2max_ (mL·kg^−1^·min^−1^)	Push-Ups (reps)	Sit-Ups (reps)	Bench Press (reps)	Sit and Reach (cm)	Power (in)	Overall Fitness
BMI	−0.605 **	−0.375 **	−0.288 **	0.249 *	−0.136	−0.290 **	−0.291 **

* *p* ≤ 0.05 level (2 tailed); ** *p* ≤ 0.01 level (2 tailed).

**Table 8 ijerph-18-11733-t008:** Relationship between Fitness and BMI at Week 7.

Variables	Estimated VO_2max_ (mL·kg^−1^·min^−1^)	Push-Ups (reps)	Sit-Ups (reps)	Bench Press (reps)	Sit and Reach (cm)	Power (in)	Overall Fitness	Fitness	FFAbility
BMI	−0.578 **	−0.400 **	−0.208	0.303 **	−0.057	−0.269 *	−0.265 *	−0.032	−0.176

* *p* ≤ 0.05 level (2 tailed); ** *p* ≤ 0.01 level (2 tailed).

**Table 9 ijerph-18-11733-t009:** Week 1 Hierarchal Regression.

Model	R^2^	Adjusted R^2^	R^2^	F	*p*-Value
Level 1: Age, BMI	0.03	0.01	0.03	1.54	0.22
Level 2: Cardiovascular Endurance	0.26	0.23	0.22	25.75	≤0.001
Level 3: Muscular Endurance	0.45	0.41	0.19	9.43	≤0.001
Level 4: Power	0.46	0.41	0.01	1.23	0.27

Level 1: Predictors: BMI, Age, Level 2: Predictors: BMI, Age, Est-VO_2max_, Level 3: Predictors: BMI, Age, Est-VO_2max_, Bench press, push-ups, sit-ups, Level 4: Predictors: BMI, Age, Est-VO_2max_, Bench press, push-ups, sit-ups, power.

**Table 10 ijerph-18-11733-t010:** Week 7 Hierarchal Regression.

Model	R^2^	Adjusted R^2^	R^2^	F	*p*-Value
Level 1: Age, BMI	0.01	−0.02	0.01	0.21	0.81
Level 2: Cardiovascular Endurance	0.40	0.38	0.39	55.53	≤0.001
Level 3: Muscular Endurance	0.46	0.42	0.06	3.18	≤0.05
Level 4: Power	0.46	0.41	0.00	0.03	0.87

Level 1: Predictors: BMI, Age, Level 2: Predictors: BMI, Age, Est-VO_2max_, Level 3: Predictors: BMI, Age, Est-VO_2max_, Bench press, push-ups, sit-ups, Level 4: Predictors: BMI, Age, Est-VO_2max_, Bench press, push-ups, sit-ups, power.

**Table 11 ijerph-18-11733-t011:** Week 7 Fitness Hierarchal Regression.

Model	R^2^	Adjusted R^2^	R^2^	F	*p*-Value
Level 1: Age, BMI	0.06	0.04	0.06	2.92	0.06
Level 2: ΔFitness	0.12	0.09	0.06	5.74	0.02

Level 1: Predictors: BMI, Age, Level 2: Predictors: BMI, Age, Fitness.

## Data Availability

Data from this study are safely stored in the Exercise Psychophysiology Laboratory (ExPPL) at the University Illinois Urbana-Champaign.

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
