# Peer review of "Fitness Fights Fires: Exploring the Relationship between Physical Fitness and Firefighter Ability"

_ijerph, 2021, doi:10.3390/ijerph182211733_

Round 1

Reviewer 1 Report

Overall: This study provides novel insight into predictors of firefighting performance among firefighter recruits in a training academy. Due to previous literature being cross-sectional in nature, having both fitness and performance data at the beginning and end of a training academy in the current study is novel as it allows for the examination of how changes in fitness may influence changes in firefighting performance. However, the manuscript is lacking a clear purpose in its current form. Notably, it is not clear why cross-sectional analyses were performed at both Week 1 and Week 7. In addition, the decision-making regarding how the hierarchical regression models were constructed (i.e., order of variables entered) and how 'ΔFitness' was calculated is unclear. It seems as if the authors would be better served simply calculating a Week 1 to Week 7 change (Δ) metric for each physical fitness measure and examining how changes in these metrics predict Δperformance (and/or Week 7 performance after control for Week 1 performance). Regardless, clearly articulating the purpose(s) of each analysis would improve the clarity of the manuscript and strengthen the interpretation(s) of the results.

Line 12 (and beyond): Please provide the meaning of 'FF' abbreviation when first using in both the abstract and the manuscript. Please also be consistent in its use throughout the manuscript (i.e., firefighter vs. firefighting). Please also make it clear that the participants are FF recruits, not actual active-duty FFs throughout.

Lines 27-32: These two sentences appears to be stating similar points - is there a way to merge and delete redundancy?

Lines 38-41: Stating that the modifiable risk factors are "neglected" is a very strong (and potentially inflammatory) statement. These factors are in fact not neglected in many departments. Rather, departments are in need of identifying methods of promoting and enhancing these factors among their members. If the authors wish to use the word "neglected", perhaps provide a citation of a paper that has identified that firefighters do indeed neglect these factors would help support this claim. Otherwise, perhaps revise the word choices.

Lines 55-57: This is a strongly worded sentence, especially considering most of the papers cited did not predict cardiovascular event incidence based on physical fitness. Please adjust the language of this sentence (and perhaps the citations) to be more accurate.

Lines 71-78: Were there significant differences in any participant characteristics between the 3 recruit class cohorts (Spring 2018, Fall 2018, Spring 2019) that potentially need to be accounted for?

Lines 82-180: Please clarify if different researchers (or the firefighter recruits themselves) conducted the various assessments and collected corresponding data. Further, since it appears more than 1 person collected these data, please provide information regarding the inter-rater reliability of each of these assessments (and provide relevant citations).

Lines 87-112: Please break into separate paragraphs based on each test protocol (push-up, sit-up, run). Please provide citations supporting these methods (if applicable).

Lines 113-131: Please break into separate paragraphs based on each test protocol (body mass, bench press flexibility, vertical jump)

Lines 126-131: Was the correction factor, previously identified in the literature, applied to the jump height? (see: McMahon JJ, Jones PA, Comfort P. Int J Sports Physiology Perform. 2016;11(4):555-557.)

Lines 179-180: What is the a priori hypothesis for predicting firefighting performance in a hierarchical manner? No rationale, nor any insight into the model building decision-making, was provided for these analyses.

Lines 195-207: Why were the variables entered into the hierarchical regression model in this order?

Lines 226-233: What is the change in fitness (Δfitness) measure? How was this calculated and what does it represent? This measure needed to be introduced in the Methods. Also, why was the change in every fitness and performance variable not calculated and examined?

References: Please ensure that improper abbreviations (i.e., FFs) are not being used in the references.

Reviewer 2 Report

This was an interesting manuscript and one that addresses a gap in the literature as it relates to preparation of future firefighters enrolled in an Academy recruit training program.  Some observations relative to the results and discussion sections of the manuscript.

  1. Please report raw values for each test in week 1 and 7.  Without this information, readers are not able place the test results in the context of previously published literature.  Specifically, push-ups, bench press, vertical jump, sit and reach, 1.5 mile time, and AFC time.
  2. Discussion, pg 7, line 270.  This is the only place that flexibility outcomes were mentioned in the manuscript.  Recent literature has examined the role of functional movement in firefighters as a better and/or different way to capture flexibility.  The discussion would benefit from a brief discussion of the sit and reach outcomes as it relates to movement screens in firefighters.  Examples and references within include: Bock et al (2015). J. Mil. Veterans Health 23:33.; Cornell et al (2021) Int. J. Environ. Res. Public Health 18: 365.
  3. Page 9, line 349:  sentence indicates bench press test used an 85 lb barbell (also mis-spelled), but the methods page 3, line 119 indicated it was an 80 lb barbell.  Please correct the one that is incorrect.
  4. Pg 9, line 365.  "wether" should be "whether".
  5. pg 9, line 371:  There is literature that provides prediction equations for the YMCA Bench Press test to predict max strength.  See Kim (2002). Journal of Strength and Conditioning Research, 16(3): 440-445.
  6. page 9, line 363-369: please provide information in the methods or discuss here about what kind of fitness training occurred in the 7-week Academy.  That will help understand the suggestions of this paragraph.
  7. Some discussion about the 7 week length of the Academy would strengthen the relationship.  Were any pairwise comparisons conducted to determine of there was a significant change in the pre to post results?  If there was no significant change, then the regression using a change score may need to be re-considered.  Training literature and prior firefighter recruit studies may suggest that some of these measures may or may not change in a 7 week time period,

Round 2

Reviewer 1 Report

Overall: The purpose and analytical decision making of the manuscript has been greatly improved. As a result, the readability and impact of the manuscript has been strengthened tremendously. Only minor comments are denoted below.

Line 83: Please correct the typo of "Weel-1" to "Week-1".

Lines 97-99: Please include a citation for the ACSM fitness classifications.

Lines 444-456: This added section appears to be information better suited for the Methods section when describing the firefighter recruit academy. Please consider moving.
